# Liquid Water Content in Ice Estimated Through a Full-Depth Ground Radar Profile and Borehole Measurements in Western Greenland

Joel Brown [1,2], Joel Harper [2], Neil Humphrey [3]

[1]Aesir Consulting LLC, Missoula, Montana, 59801, USA
[2]Department of Geosciences, University of Montana, Missoula, Montana, 59801, USA
[3]Department of Geology and Geophysics, University of Wyoming, Laramie, Wyoming, 82071, USA

*Correspondence to*: Joel Brown (joel@aesirmt.com)

**Abstract.** Liquid water content (wetness) within glacier ice is known to strongly control ice viscosity and ice deformation processes. Little is known about wetness of ice on the outer flanks of the Greenland ice sheet, where a temperate layer of basal ice exists. This study integrates borehole and radar surveys collected in June, 2012 to provide direct estimates of englacial ice wetness in the ablation zone of western Greenland. We estimate electromagnetic propagation velocity of the ice body by inverting reflection traveltimes from radar data. Our inversion is constrained by ice thickness measured in boreholes and by

positioning of a temperate/cold ice boundary identified in boreholes. Electromagnetic propagation velocities are consistent with a depth-averaged wetness of ~0.5-1.1%. The inversion indicates that wetness within the ice varies from <0.1% in an upper cold layer to ~2.9-4.6% in a 130-150 m thick temperate layer located above the glacier bed. Such high wetness should yield high rates of shear strain which need to be accounted for in glacial flow models that focus on the ablation zone of Greenland. This high wetness also needs to be accounted for when determining ice thickness from radar measurements.

## 1 Introduction

As ice flows outward from the centre of Greenland Ice Sheet (GrIS), heat is added at the base of the ice column. This heat is due to geothermal heat flux from the bed, strain heating in ice, and basal friction in places where sliding occurs. Several ice sheet flow models depict that in western Greenland enough heat is eventually added to develop a fully temperate layer of basal ice (e.g.; Brinkerhof et al., 2011; Lüthi et al., 2015; Meierbachtol et al., 2015), however substantial modelling uncertainty

exists regarding the spatial extent and vertical dimensions of the warm basal layer. Limited borehole temperature observations have confirmed a temperate basal layer in western Greenland that is tens of metres to well over 100 meters thick (Lüthi et al., 2002; Ryser et al., 2014; Harrington et al., 2015). This layer potentially plays a key role in deformational motion of the ice sheet.

Measurement of the GrIS's surface motion has become relatively commonplace, and basal sliding speed can be partitioned

from surface velocity as the residual after modelled deformational velocity has been removed. An accurate representation of

ice viscosity is a fundamental ingredient for reliable numerical simulations of deformational velocity. The viscosity of ice at the melting point is highly dependent on the liquid water fraction (wetness) of the ice (Duval, 1977). Incorporation of wetness properties into numerical models can therefore result in substantial modifications to simulation output (e.g., Greve, 1997; Hubbard et al., 2003; Aschwanden et al., 2012). Unfortunately, due to a lack of observational evidence for constraining the

water content of the ice, poorly constrained modelling decisions must be made regarding the water content and therefore viscosity of the GrIS temperate basal layer.

Direct measurement of in situ ice wetness is not straightforward, as field samples would somehow need to be collected and then measured in an unaltered condition. Several studies (conducted outside of Greenland) have therefore utilized Ground Penetrating Radar's (GPR) strongly sensitive propagation velocity to volumetric water content to make non-invasive estimates

of englacial water content (e.g., Macheret et al., 1993; Bradford and Harper, 2005; Murray et al., 2007; Bradford et al., 2009). A drawback of radar-based studies is that the water in grain-scale and macro-scale inclusions cannot be distinguished or partitioned since the propagation speed of the radar is dependent on the bulk properties of the ice/water mixture. In addition, measuring radar propagation velocity in a thick ice sheet setting such as Greenland requires a cumbersome data collection scheme, with large and variable antenna separations that are not logistically attainable from airborne platforms at present.

The development of a temperate basal layer of ice along the outer flanks of the GrIS is predicted by models and confirmed by observation. Here we provide a key observation of the volumetric liquid water content of the temperate basal layer, an important constituent of the layer's rheological properties. We integrate multiple datasets including: (1) direct measurement of ice thickness and ice temperature from instrumented boreholes extending to the bed of the ice sheet; (2) common offset GPR transects acquired with 3 different frequencies; and (3) a common-source point multi-offset GPR survey (equivalent

geometry to a single shot gather in seismic data). Our purpose is to quantify the liquid water content of a basal temperate layer spanning a reach between boreholes by employing a geophysical inversion of the GPR data with constraints from borehole data.

## 2 Methods

### 2.1 Field Site and Boreholes

The study was conducted in western Greenland, ~27 km inland from the terminus of Isunngua Sermia, a land terminating outlet glacier on the western side of the GrIS (Fig. 1a). The study site is within the ablation zone and located at approximately 820 – 850 m elevation. Using a hot water drill, we drilled a total of 7 boreholes to the bed at two locations separated by ~1 km. Three holes separated by <30 m were drilled in 2011 at S3, and four holes separated by <50 m were drilled in 2012 at S4 (Fig. 1b). Sensor strings with various instruments including thermistors spaced on 20 m intervals were deployed into the boreholes

and allowed to freeze into the ice and equilibrate with the ambient ice temperature (Harrington et al., 2015).

The three boreholes drilled at S3 had an average depth of 461 m with a standard deviation of 4 m, and the four boreholes drilled at S4 had an average depth of 698 m with a standard deviation of 8 m (Table S1, Supplementary Information). The

GPR transect between borehole sites has less than 30 m of surface elevation difference, but more than 200 m of bed elevation change as it extends from a relative high ridge to the centre of a bed trough (Fig. 2). The radar transects start at S3 and run in a nearly straight line to within 120 m of S4 where we encountered a large, deeply incised stream which we could not cross (Fig. 1b). Temperature data were recorded with a string of temperature-sensing semiconductor chips spaced at 20 m intervals

from the ice surface. Data were retrieved from the chips via laptop computer 3 months after the temperature strings were placed in the boreholes, allowing time for the chips to equilibrate to the ice temperature. Thus, measured temperatures are representative of the time of retrieval. Temperature profiles from both borehole sites indicate the ice near the surface is about -6 °C due to the winter cold wave, and below that is about -4 °C until a boundary that separates the upper cold ice and temperate basal ice (Fig. 3c). Below the boundary, the ice is entirely temperate, occupying a layer ~130-150 ± 10 m thick at both S3

and S4 or roughly 30% and 20% of the entire ice thickness, respectively.

## 2.2 Ground Penetrating Radar surveys

We collected three common offset Ground Penetrating Radar (GPR) profiles and a common-source point multi-offset GPR survey between borehole sites S3 and S4. Data were collected using a custom GPR system deployed on the ice surface. The transmitter consisted of a Kentech pulser which sends a ±2 kV electrical pulse into a resistively loaded antenna with a temporal

frequency of 1 kHz. The receiver consisted of a resistively loaded antenna attached to a Pico 4227 oscilloscope; we controlled the oscilloscope and recorded data via a MATLAB script. The oscilloscope was triggered by the arrival of the airwave on one channel and data were recorded on a second channel. The two-way traveltime of the wave is adjusted to account for traveltime of the airwave between transmitting and receiving antennas for all surveys. The common offset radar survey was repeated with three different length antenna sets with nominal centre frequencies in ice of 2.5 MHz, 5 MHz, and 10 MHz. We collected

common offset transect GPR data in TE (transverse electric) mode and common-source point multi-offset survey data in TM (transverse magnetic) mode.

Bradford et al. (2013) point out that the polarity mode of the GPR survey can have large effects on the propagation velocity of the EM wave in wet, fractured ice. By comparing the nearest offset common source trace (TM mode) to the coincident 2.5 MHz common offset trace (TE mode) we find no significant discrepancy in the traveltime to the bed or to the internal reflection

at 3584 ns; from this we conclude that fracture alignment is not greatly influencing our measurements.

Present at the surface during data collection were ridges up to ~2 m in height related to ice foliation and melt processes, active streams either deeply incised or occupying flat-bottomed linear troughs also related to ice foliation, pools of liquid water, and thousands of water-filled cryoconite holes (Fig. 1b). The filters applied to data include a high-pass filter with a cut-off frequency of one-half the peak frequency (dewow) to reduce very low-frequency noise, a time variable gain to account for

spherical spreading and attenuation of the signal, and an Ormsby bandpass filter to reduce high and low frequency noise. Survey and filter settings are shown in Table 1 for each GPR survey.

Common offset data were acquired by manually stepping both antennae forward while maintaining a constant spacing between the transmitting and receiving antennas (antenna offset). The antennas were held stationary for the duration of acquisition for

each trace which included stacking 128 traces at each location. Although this resulted in a time-consuming survey, we successfully avoided spatial aliasing while allowing for enough stacking to achieve a signal-to-noise ratio high enough to allow interpretation of the data. For the common-source point multi-offset GPR survey, the transmitting antenna was left stationary near S3 for the duration of the acquisition as the receiving antenna was manually stepped toward S4. As the receiving antenna was stepped away from the transmitting antenna, the recorded signal strength diminished as a function of the spherical spreading and signal attenuation. Thus, the threshold level for triggering was manually reduced with greater offset. The amplitude of each trace was also normalized to the airwave during processing of the common-source point profile to account for system variability that accompanied the reduction of the triggering threshold.

The common-source point multi-offset survey that we use in this study is preferable to common midpoint (CMP) surveys for regions where reflecting surfaces, including internal layering and the bed, are not planar surfaces. This is because CMP surveys assume that the point of reflection for each offset is invariant, thus residual moveout correction must be employed in regions where the reflecting surfaces are not parallel with the survey surface. This, in turn, requires a priori knowledge of the magnitude of the dip angle for all reflecting surfaces. However, the survey setup and inversion used in this study do not require residual moveout correction or a priori knowledge of the magnitude of the dip angle since the dip of reflecting surfaces is solved for in the inversion. Further, as described in Brown et al. (2012), the inversion used in this analysis is not subject to normal moveout (NMO) assumptions such as small offset to depth ratios and small velocity gradients over reflection boundaries; velocity/depth models derived from CMPs are most commonly determined through semblance analysis and the Dix inversion (Dix, 1955) of solving for layer velocities, which *is* subject to NMO constraints as well as small errors in NMO velocities and near-offset traveltime picks.

**2.3 EM Velocity**

Average electromagnetic (EM) propagation velocity of the glacial ice was measured using two separate methods.

1) A direct comparison between the two-way traveltime of the GPR bed reflection and the measured depths of the boreholes. We calculate the average EM propagation velocity at each borehole location with a simple two-way traveltime vs. depth relationship Eq. (1)

$$v = \frac{2d}{TWT},\tag{1}$$

Here, $v$ is the EM propagation velocity, $d$ is the ice thickness, and *TWT* is the Two-Way Traveltime of bed reflection. While this method is the most straightforward and accurate, it is rarely employed as boreholes are not typically available.

2) A ray-based traveltime inversion (Zelt and Smith, 1992) of the common-source point multi-offset data is constrained by the direct moveout of the surface wave and borehole temperature and depth measurements. This inversion employs a forward raytracing model to determine the traveltime of a reflected wave from the transmitter to the receiver based on an input velocity/depth model; in each iteration of the inversion, the velocity/depth model is adjusted using a dampened least squares method. We solved the ray-based traveltime inversion with a two-layer model which allows us to constrain vertical variations

in the GPR propagation velocity structure of the ice column. This inversion technique employs a model space that mimics the ice thickness at our field site, which has a nearly flat surface on the scale of the 2.5 MHz GPR and a bed dip angle that exceeds 14° in some locations. The ray-based traveltime inversion directly mimics the survey geometry, where the transmitting antenna is stationary and the receiving antenna is moved linearly away from the transmitter on spatially constant intervals.

The ray-based traveltime inversion requires an initial 2D velocity model which we constrained with the apparent bed geometry derived from our common offset radar profiles, the measured borehole depths, and the boundary between the upper, cold ice layer and the lower, temperate ice layer which was measured with the borehole derived vertical temperature profiles at S3. The input velocity model geometry is based on unmigrated common offset GPR data since migration of the data does not affect the geometry of the profiles in the region where bed reflections are modelled (see supplementary information). We used the

interfacial surface wave velocity measured with the common-source point multi-offset survey to further constrain our traveltime inversion.

The initial velocity model consists of two layers with homogenous velocity distribution. We use the deepest continuous layer observed in the common offset GPR data to extend the cold/temperate boundary toward S4 in our initial velocity model. Using this layer as an initial model boundary is arbitrary to an extent, however, we chose this layer as our model boundary since this

layer is coincident with the temperate/cold ice boundary observed in the borehole temperature profile at S3 when the depth is calculated using the measured surface wave velocity. This does not indicate that the reflection is due to a boundary between cold and temperate ice or that the boundary between cold and temperate ice is coincident across the profile. Thus, the depth of the modelled cold/temperate layer boundary is allowed to vary in our inversion in places where direct measurements of temperature do not exist. We employ the traveltime inversion by solving for the depth of the upper layer while holding the

velocity constant before solving for the bulk velocity of the lower layer and the depth to the bed. Our traveltime inversion solution is further constrained by the average EM propagation velocity of the ice column as measured at the boreholes and the measured thickness of the glacier at the two borehole sites at the ends of the transect.

## 2.4 Ice Water Content

Electromagnetic wave propagation velocity through wet ice is dependent on many physical properties of the medium including

the size, shape, and orientation of the water bodies within the ice as well as impurity concentration of both ice and water, unfilled porosity (air bubbles trapped within ice), frequency range of the EM signal, and the polarity mode of the survey. There are several mixing models that approximate the relationship between EM propagation velocity including the Looyenga (1965) mixing model and the complex refractive index model (CRIM) (Wharton et al., 1980). Both of these mixing models have been used to approximate the wetness of glacial ice in previous studies with reasonable results (e.g., Bradford and Harper, 2005;

Murray et al., 2007). Herein we estimate wetness calculated from the 2 phase form of the both the CRIM equation as well as the Looyenga (1965) equation. To apply these equations, we assume that (1) randomly aligned cold glacial ice has an EM phase velocity of $1.685 \times 10^8$ m s$^{-1}$ $\pm$ 0.5% ($1.68 \times 10^8$ m s$^{-1}$ to $1.695 \times 10^8$ m s$^{-1}$) (Fujita et al., 2000), (2) large voids within the ice may be filled with incompressible water, but there are no air filled voids, and (3) conductivity is negligible. It is implicit

in both these equations that the wetness, or liquid water content, is given as a volumetric percentage. Thus, all of the wetness results based on these equations are given as volumetric percentages.

The 2 phase CRIM equation for water inclusions in ice is Eq. (2)

$$\theta_w = \frac{c/v - \sqrt{K_i}}{\sqrt{K_w} - \sqrt{K_i}},\tag{2}$$

where $\theta_w$ is the volume percentage of water, $c$ is the speed of light in free space ($\sim 3 \times 10^8$ m s$^{-1}$), $v$ is the measured propagation velocity in the ice, $K_i$ is the relative dielectric permittivity of glacial ice ($\sim 3.17$), and $K_w$ is the relative dielectric permittivity of water ($\sim 86$). Implicit in this equation is the relationship between EM propagation velocity and the relative dielectric permittivity of a medium with negligible conductivity ($K = (c/v)^2$).

For wet ice, the Looyenga (1965) equation takes the form Eq. (3)

$$\theta_w = \frac{(c/v)^{2/3} - K_i^{1/3}}{K_w^{1/3} - K_i^{1/3}}.\tag{3}$$

We report the wetness values derived from both equations in Table 2, but we restrict our discussion to the results from the 2 phase CRIM equation since it yields more conservative values for liquid water content.

# 3 Results

## 3.1 Constant offset reflectors

The bed reflection is imaged in common offset profiles at all three frequencies (Fig. 2); it is most apparent in the 2.5 MHz data and most precise in the 10 MHz data. For this reason, we use the 10 MHz data to determine the TWT of the bed reflection. These data show that the TWT to the bed reflection increases from S3 to S4 by $2.75 \times 10^{-6}$ s $\pm$ $0.14 \times 10^{-6}$ s. This agrees with the $\sim 240$ m $\pm$ 15 m increase in ice thickness measured in boreholes. The apparent dip angle of the bed is $\sim 18°$ between 300 m and 600 m distance along the unmigrated transect (Fig. 2). In this same region, the amplitude of the bed reflection observed in the 10 MHz data is similar to the noise level of the data and it can only be traced because it is laterally coherent. The bed reflection seen in the 5 MHz data at 600 m distance appears to be discontinuous; this is likely due to off nadir reflections from a much steeper bed slope in this region.

All of the common offset GPR profiles also imaged internal layering throughout the ice body, although the signal to noise ratio of the deeper layers diminished at higher frequencies. It is apparent in the 10 MHz profile that the internal reflection horizons within the upper 30 % of the ice all approximately double in depth from S3 to S4. The internal reflectors in the lower 70 % of the ice, including the lowest continuous reflection, increase in depth by $\sim 60$ %. Hence, surface layers are slightly thicker over the deep trough compared to the thinner ice. In all three profiles, reflections from point sources with lateral coherence of less than 100 m were imaged between 400 ns and 3800 ns TWT (Fig. 2). These features where not widespread, but were restricted to locations within 400 m of S3. It is possible that these internal reflections are from off-nadir sources, however, they are neither from near surface sources such as crevasses, moulins, cracks or pools of water, nor from off nadir bed topography.

Analysis of our data in conjunction with two Operation IceBridge MCoRDS L1B Geolocated Radar Echo Strength Profiles (Leuschen, 2014) which cross the common offset radar profile presented here reveal that there are no large variations in the out-of-plane bed geometry (Fig. 4). The interpreted bed reflection in the IceBridge data nearest S3 is inconsistent with both the measured borehole depths at S3 as well as the depth of the bed we interpret in our radar data; the IceBridge interpreted ice thickness is ~80 m to 90 m less than the actual ice thickness confirmed by drilling at S3 (Fig. 4b). This may indicate that the higher frequency MCoRDS radar (between 180 MHz and 210 MHz) does not penetrate the higher water content of the basal temperate layer with high enough amplitude to resolve the bed reflection.

In all three common offset profiles, various unidentified sources of noise and occasional errors in timing due to difficulties in triggering off of the airwave created both coherent and non-coherent noise in the data. The most apparent noise recorded in our surveys were coherent 'ghost' signals observed in the 5 MHz and 2.5 MHz data (Fig. 2b&c). The source of this coherent noise is unknown, as it is not explainable by off-nadir englacial reflections, surface expressions of cracks, or surface streams or pools of water, nor is it seen in all of the radar profiles. Noise due to triggering errors is most prominent in the 5 MHz data (Fig. 2b). These noise features in the data can be ignored as they do not interfere with the interpretation of the data.

## 3.2 EM velocity and water content

### 3.2.1 Common offset GPR vs. measured borehole depths

Borehole depth measurements show that the ice thickness increases from 461 m $\pm$ 6 m at S3 to 697.5 m $\pm$ 13 m at S4, resulting in a ~51 % increase in ice thickness from S3 to S4. The GPR two way traveltimes to the bed were 5.717 x $10^{-6}$ s $\pm$ 0.092 x $10^{-6}$ s at S3 and 8.469 x $10^{-6}$ s $\pm$ 0.044 x $10^{-6}$ s at S4 (Fig. 2) resulting in calculated average EM velocities of 1.61 x $10^8$ m s$^{-1}$ $\pm$ 0.04 x $10^8$ m s$^{-1}$ at S3 and 1.65 x $10^8$ m s$^{-1}$ $\pm$ 0.02 x $10^8$ m s$^{-1}$ at S4. We use Eq. (2) to estimate the average water content of the entire ice column as 1.1 $\pm$ 0.4 % at S3 to 0.5 $\pm$ 0.1 % at S4. This is equivalent to ~5.1 m of water depth at S3 and ~3.5 m of water depth at S4.

We can further constrain this estimate by assuming that the liquid water in the ice is not uniformly distributed across the ice thickness. The surface arrival slope of the common-source point multi-offset profile shows that the EM velocity in the near surface (within the upper tens of meters) is 1.69 x $10^8$ m s$^{-1}$ $\pm$ 0.01 x $10^8$ m s$^{-1}$ (Fig. 3a). A fair assumption is that grain-scale water (or intra-granular water), liquid water occurring at the junction points between individual ice crystals (e.g.; Gusmeroli, et al., 2012), cannot exist in cold ice, meaning liquid water is concentrated in the temperate ice layer that we observe with the vertical temperature profiles (Fig. 3c) and that the measured surface wave velocity is constant through the cold layer. Accordingly, we estimate the liquid water content of the temperate layer varies between 4.6 % at S3 and 2.9 % at S4. See the Supplementary Information for the details on the calculations and discussion of errors and assumptions.

### 3.2.2 Traveltime inversion of multi-offset data

The result of the traveltime inversion reveals a bulk EM propagation velocity of $1.48 \times 10^8$ m s$^{-1}$ in the lower layer with an assumed velocity of $1.69 \times 10^8$ m s$^{-1}$ in the upper layer (Fig. 3b). This is equivalent to negligible wetness in the upper layer and 3.3 % wetness in the lower layer. The 44 ray path travel times calculated for the final velocity model match the common-source point multi-offset data with a RMS misfit of 61.1 ns. The largest single misfit within the inversion solution is 107.6 ns which is equivalent to 0.9 m vertical error, or approximately half of the height of the surface roughness over the region of data collection.

The ray-based traveltime inversion is only valid over the region where rays are present, thus much of the transect has no constraint on the depth of the temperate layer or the ice wetness. However, since the borehole temperature measurements reveal that the temperate layer is the same thickness at both S3 and S4, we infer that the temperate layer thickness is fairly uniform across the profile.

## 4 Discussion

### 4.1 Comparison of results

The temperate basal layer present along the outer flanks of the GrIS arises from geothermal heating and flow mechanisms acting on cold ice moving outward from centre of the ice sheet. These warming processes differ from other temperate glacier settings where ice is warmed to the melting point during ice diagenesis related to surface melt and infiltration. This begs the question of how the water content of the warm basal layer in Greenland differs from other measurements of temperate ice. Various non-radar based methods for estimating ice water content have suggested liquid water content of temperate glaciers ranging from 0.0 % to 3.0 % (e.g.; Raymond and Harrison, 1975; Vallon et al., 1976) . Studies employing GPR propagation velocity for wetness measurement have reported liquid water content values ranging from 0.0 % to 7.6 % (e.g.; Macheret et al., 1993; Moore et al., 1999; Murray et al., 2000; Gusmeroli et al., 2012). Although values of up to 9.1 % have been reported (Macheret and Glazovsky, 2000), most measurements indicate wetness of less than 4 %. Our results of 2.9 to 4.6 % for the mean wetness across the 130-150 m thick basal warm layer are thus slightly high end, but not out of range compared to other estimates for temperate ice.

Our results are higher than the estimate by Lüthi et al. (2002) of 2 % water content at cold ice/temperate layer interface at a site in the Jakobshavn region of Greenland. The latter was based on an observationally constrained model of refreezing at the layer boundary, whereas ours are averaged over the full thickness. Whether the higher water content we observe in the temperate layer represents grain-scale water or a mix of grain-scale and macro-scale bodies, is not known. Harrington et al. (2015) documented vertical growth of the temperate layer along a flow line transect. They argued that the only mechanism for expanding the temperate layer vertically was through basal crevassing. If this is indeed the case, an important fraction of our

measured water content is likely located in macro scale basal crevasses. Our radar has not provided obvious imaging of any such crevasses, but our methods were not targeted to their detection.

The water content we observe in Greenland's temperate basal layer is substantially higher than the 1 % typically employed as a maximum cut-off in ice sheet models (e.g.; Greve, 1997; Aschwanden et al., 2012). A drainage function is applied at this threshold to represent drainage of water in excess of 1 % through crevasses, cracks, and grain boundaries (Greve, 1997). This conceptualization, however, is limited to the grain-scale water content and has no accommodation of water storage in drainage features, whereas our results represent liquid water potentially existing at all scales.

## 4.2 Where the water is being held within the ice

While the cold layer may not have grain-scale water our common offset imaging of englacial hyperbolas imply macro-scale water bodies are present in this layer. Point reflectors in similar data have been interpreted either as near surface crevasses (presumably the same as water filled voids) or surface crevasses that are off-axis from the radar profile (Catania et al., 2008). The hyperbolic diffractions in our common offset data are unlikely to result from a distant near-surface source, since the theoretical and measured radiation pattern of a dipole antenna (Arcone, 1995) greatly limits this possibility: the relative signal strength of off nadir, near surface reflections would be weak compared to reflections generated at nadir, and we do not observe that. Rather, the diffractions observed in our profiles are strong, indicating that they likely arise from near-nadir discontinuities. Similar hyperbolic returns in the cold ice layer of polythermal glaciers have been observed prior to any seasonal melt, and have been interpreted as meter scale water bodies persisting through the winter (e.g.; Pälli et al., 2002). Therefore, our working hypothesis is that the cold layer contains sparse large water inclusions up to several hundred meters below the surface, perhaps generated in a crevassed area about 3 km up flow from the site. Further, since our estimates of liquid water content are derived under the assumption of negligible water content in the upper layer, our wetness values for the temperate layer would be slightly high if the englacial bodies that produce the point reflections seen in the common offset radar data contain non-negligible volumes of liquid water.

## 4.3 Implications for creep rate

Warm ice near the bed requires a lower activation energy than cold ice to initiate creep of the polycrystalline structure. Furthermore, laboratory measurements reveal that the strain-rate of ice triples as the grain scale wetness increases from relatively dry conditions of 0.01 % to the modestly wet value of 0.8 % (Duval, 1977). That all of the water we observe is in macro scale features and none exists at the grain scale is unlikely since water is known to accumulate at grain boundaries in temperate ice (e.g.; Shreve et al., 1970; Nye and Frank, 1973). It follows that the temperate layer should contain sufficient grain scale liquid water to permit enhanced strain rates. Further, the impact of water located in macro scale features such as basal crevasses on ice rheology has not been quantified, but is also likely to soften the ice.

With a soft basal layer, partitioning of surface velocity into sliding and deformation components would attribute enhanced deformation in the temperate layer to basal sliding processes, unless the enhanced creep is explicitly accounted for. Borehole observations have attributed 44 - 73 % of winter motion to basal sliding (Lüthi et al., 2002; Ryser et al., 2014), however, tilt sensors do not freeze in place within the temperate layer to yield reliable readings, thus making the distinction between motion

due to high straining of the temperate layer and motion due to sliding processes ambiguous.

## 4.4 Implications for ice thickness

A final implication of our results relate to Greenland's ice thickness and volume. Airborne radar has been deployed for decades to image the GrIS's internal layers and bedrock topography, and these data have been used to generate high resolution digital elevation models of the ice sheet bed (e.g.; Bamber et al., 2013). Depth conversion of the airborne radar data typically employs

Eq. (1) with assumptions about the radar velocity, namely that the velocity is constant and the conversion from TWT to depth assumes a bulk permittivity of 3.15. Our direct comparisons of TWT to borehole depths demonstrate the radar propagation speed in this region of the GrIS is influenced by liquid water in the basal temperate layer of ice. Neglecting the temperate layer and solving Eq. (1) using just the propagation speed of cold ice could overestimate ice thickness by 20 m at S3 and 15 m at S4. This would be equivalent to 4.3 - 2.1 % overestimate of ice thickness at S3 and S4, respectively. In our study area we find

potential error due applying Eq. (1) with a fixed cold-ice velocity would scale inversely with thickness of the ice since the basal temperate layer has constant thickness. However, in practice we found the opposite occurred—that the ice thickness was underestimated in the interpretation of the airborne radar data. For example, near S3 the IceBridge data show a TWT of 4.4545 x $10^{-6}$ s between the surface and interpreted bed reflections (Leuschen, 2014), and assuming cold ice propagation velocity this results in an estimated depth of ~375 m which underestimates ice thickness by about 80 m (~18 %) when compared to measured

borehole depths at S3. This discrepancy could have at least three causes: 1) the bed picking algorithm yields a smoothed representation of bed roughness, whereas borehole depths capture local variability; 2) internal reflectors mask the bed reflection such that the bed is poorly recognized; and, 3) the contrast in bulk dielectric permittivity due to the high water content of the temperate basal ice produce a reflection surface which was interpreted to be the bed. Errors in thickness due to liquid water in the temperate layer likely vary substantially around the ice sheet, but clearly do not necessarily scale directly with ice thickness

everywhere. Such errors may be second order in many uses of the data, but may also be quite relevant to others, such as the propagating assumptions taking place in calculations of mass conserving beds (e.g., Morlighem et al., 2013).

## 5 Conclusions

Our integration of ground based-radar data with information collected in boreholes reveals a two-layer, thermo-hydrologic structure of varying thicknesses in the ablation zone of western Greenland. Our results are based on study of a ~1 km long

transect, located 26 km inland from the ice terminus, and constrained with 3 boreholes to ~460 m at one end and 4 boreholes

to ~700 m at the other end. We find consistent results from radar surveys collected at three different frequencies and with different survey configurations and methods for radar velocity inversion of liquid water content.

The ice mass is best described as a two-layer stratified system, with a cold upper layer overlying a warm temperate layer above the bed. The boundary between the two layers corresponds to a thermal transition from cold ice to temperate ice having liquid water. The temperate layer maintains nearly constant thickness of about 130 - 150 m as the total ice thickness increases by 50 % over a bedrock trough. The cold layer contains rare point reflectors hundreds of meters below the surface, which are likely water filled and perhaps generated in an icefall above the study reach. If we further assume that upper, cold ice layer has negligible wetness, the temperate basal layer has a water content of 2.9 - 4.6 %. This range is substantially higher than the cut-off value typically used in ice sheet models, although the fraction of our measured range located in macro-scale features such as basal crevasses is unknown.

**Acknowledgements**

This work is funded by SKB-Posiva-NWMO through the Greenland Analogue Project and NSF (Office of Polar Programs–Arctic Natural Sciences grant no. 0909495). We thank Dr. Joseph MacGregor, Dr. Achim Heilig, and two anonymous reviewers for their comments which greatly improved the manuscript.

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

| Survey type | Antenna Length (m) | Frequency (MHz) | Estimated wavelength (m) | Antenna offset (m) | Trace separation (m) | Gain | Bandpass cutoff frequencies (MHz) |
|---|---|---|---|---|---|---|---|
| Common offset | 40 | 2.5 | 65.2 | 40 | 10 | $t^{1.2}$ | 0, 2, 5, 8 |
| | 20 | 5 | 36.6 | 20 | 5 | $t^{0.9}$ | 2, 5, 7, 10 |
| | 10 | 10 | 16.3 | 10 | 2.5 | $t^{0.8}$ | 0.5, 7, 15, 17 |
| Common-source point | 40 | 2.5 | 65.2 | 40 | 10 | $t^{1.2}$ | Not applied |

**Table 1. Acquisition parameters for GPR surveys collected for this study.**

| EM velocity calculation method | | S3 | | S4 | |
|---|---|---|---|---|---|
| Direct TWT to borehole depth (Bulk average) | EM velocity (m s$^{-1}$) | $1.61 \times 10^8 \pm 0.03 \times 10^8$ | | $1.65 \times 10^8 \pm 0.01 \times 10^8$ | |
| | CRIM wetness (%) | $1.1 \pm 0.4$ | | $0.5 \pm 0.1$ | |
| | Looyenga (1965) wetness (%) | $1.5 \pm 0.6$ | | $0.7 \pm 0.2$ | |
| Direct TWT to borehole depth (Two layer) | EM velocity (m s$^{-1}$) | upper layer $1.69 \times 10^8$ | lower layer $1.41 \times 10^8 \pm 0.11 \times 10^8$ | upper layer $1.69 \times 10^8$ | lower layer $1.50 \times 10^8 \pm 0.06 \times 10^8$ |
| | CRIM wetness (%) | upper layer $< 0.1$ | lower layer $4.6 \pm 2.9$ | upper layer $< 0.1$ | $2.9 \pm 0.9$ |
| | Looyenga (1965) wetness (%) | upper layer $< 0.1$ | lower layer $6.3 \pm 3.7$ | upper layer $< 0.1$ | lower layer $4.0 \pm 1.2$ |
| Two layer traveltime inversion | EM velocity (m s$^{-1}$) | upper layer $1.69 \times 10^8$ | lower layer $1.48 \times 10^8$ | Not applicable | |
| | CRIM wetness (%) | upper layer $< 0.1$ | lower layer $3.3$ | Not applicable | |
| | Looyenga (1965) wetness (%) | upper layer $< 0.1$ | lower layer $4.5$ | Not applicable | |

Table 2. Ice wetness value results calculated at S3 and S4 using the methods described in the text.

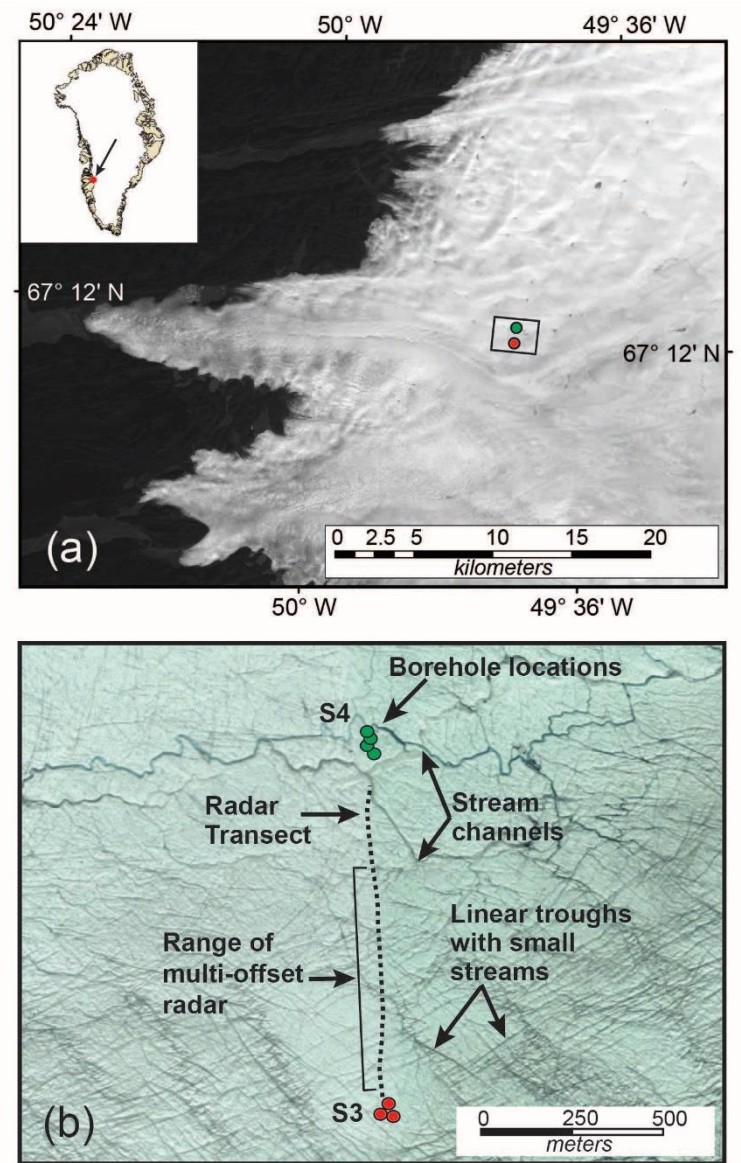

**Figure 1: (a) Landsat image showing location of borehole sites in relation to the ice sheet margin. The inset map shows the location of the study site, the rectangle shows the location of the image in panel (b). (b) Higher resolution satellite image (courtesy of Google/Digital Globe) showing the path of the radar transect (dotted line), the range of the single moveout multi-offset radar survey, and the locations of the boreholes at S3 (red dots) and S4 (green dots). It is clear that the radar transect crosses many small (<2 m) stream channels and linear water filled troughs between S3 and the deeply incised stream channel at the end of the radar transect.**

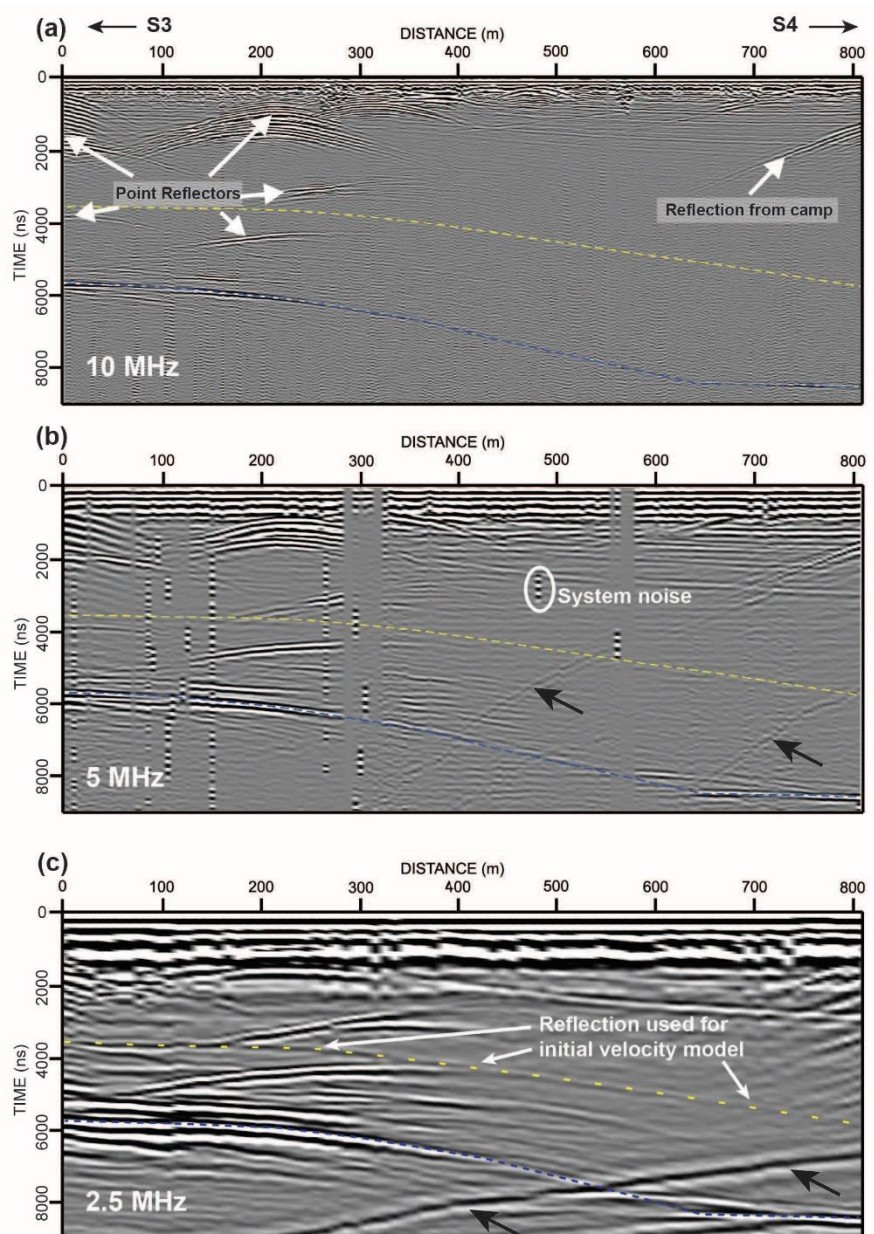

**Figure 2: Common offset radar profiles collected with 10 MHz (a), 5 MHz (b), and 2.5 MHz (c) antenna. The bed reflection picks (blue dotted line) are derived from the 10 MHz data and the continuous englacial reflection picks (yellow dotted line) are derived from the 2.5 MHz data. Englacial point reflectors (a) are seen in all three radar profiles within 300 m (distance) of S3. A prominent reflection from the camp (located at S4) is also present in the data (a). Englacial layering that is apparent to 250 m depth near S4, the thickness of these layers decreases towards S3. Black arrows in panels (b) and (c) show unexplained 'ghost' reflections. The 5 MHz data (b) also show the effects of system noise due to timing errors associated with improper triggering off of the airwave (white oval) as well as vertical banding in the profile around 300 m and 575 m (distance) that likely occur where antenna to surface coupling was poor. Glacial flow direction (east to west) is into the page.**

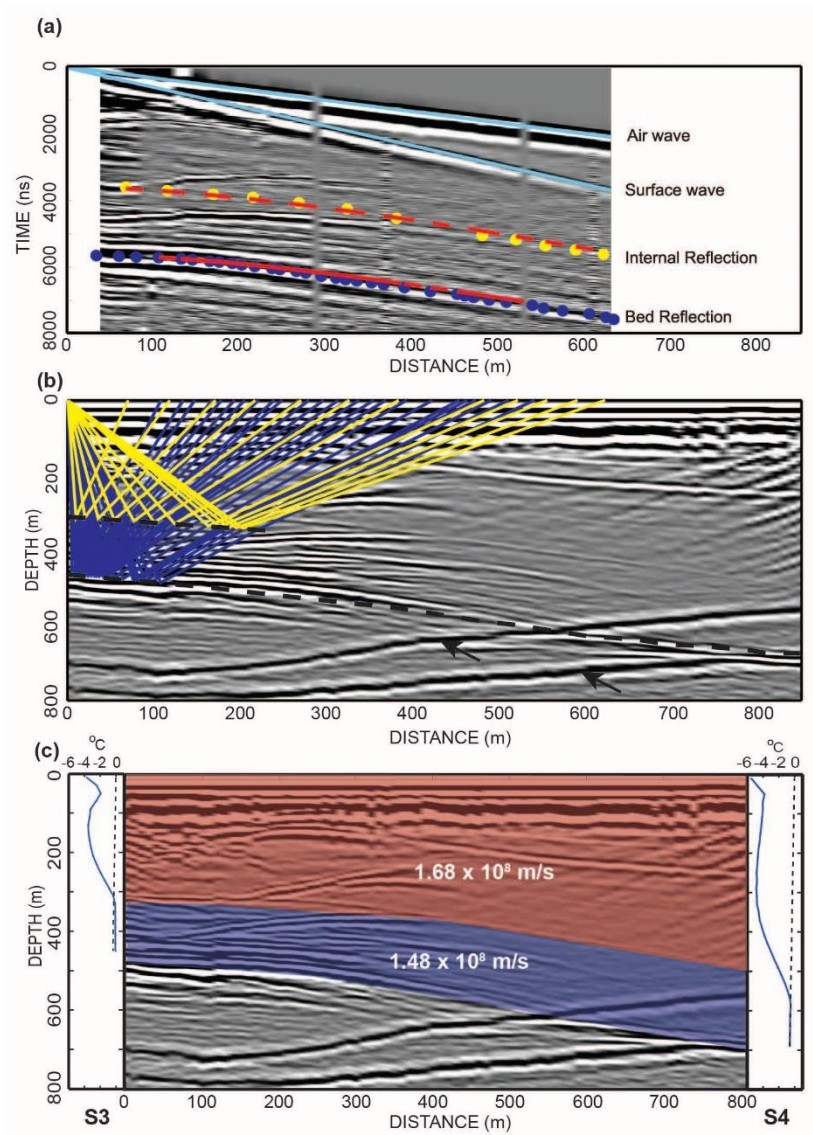

**Figure 3: (a)** Result of the traveltime inversion with the original single moveout multi-offset data image shown. The cyan lines show the linear moveout of the air wave and the direct interfacial surface wave. Blue dots are picks from the bed reflection yellow dots are picks from the internal reflection horizon shown in the common offset profiles (Fig. 2). Red dashed lines (a) are the traveltime curves calculated from the ray paths in the final traveltime model (b). The calculated ray paths (b) are coloured to show the modelled layer boundary (dashed lines in (b)) that they reflect off of, yellow ray paths reflect off of the englacial boundary between cold and temperate ice, blue ray paths reflect off of the ice/bed interface. The TWT to depth conversion of the 2.5 MHz common offset data were calculated for each trace from the final velocity model (c). The 'ghost' reflections discussed in the text are marked with black arrows. **(c)** Vertical temperature profiles shown (blue curves) alongside the 2.5 MHz common offset profile and the ice velocity model calculated with the traveltime inversion. Black dotted lines are the estimated pressure melting point temperature profiles. The depth conversion for the 2.5 MHz common offset data are calculated with the velocity model (c).

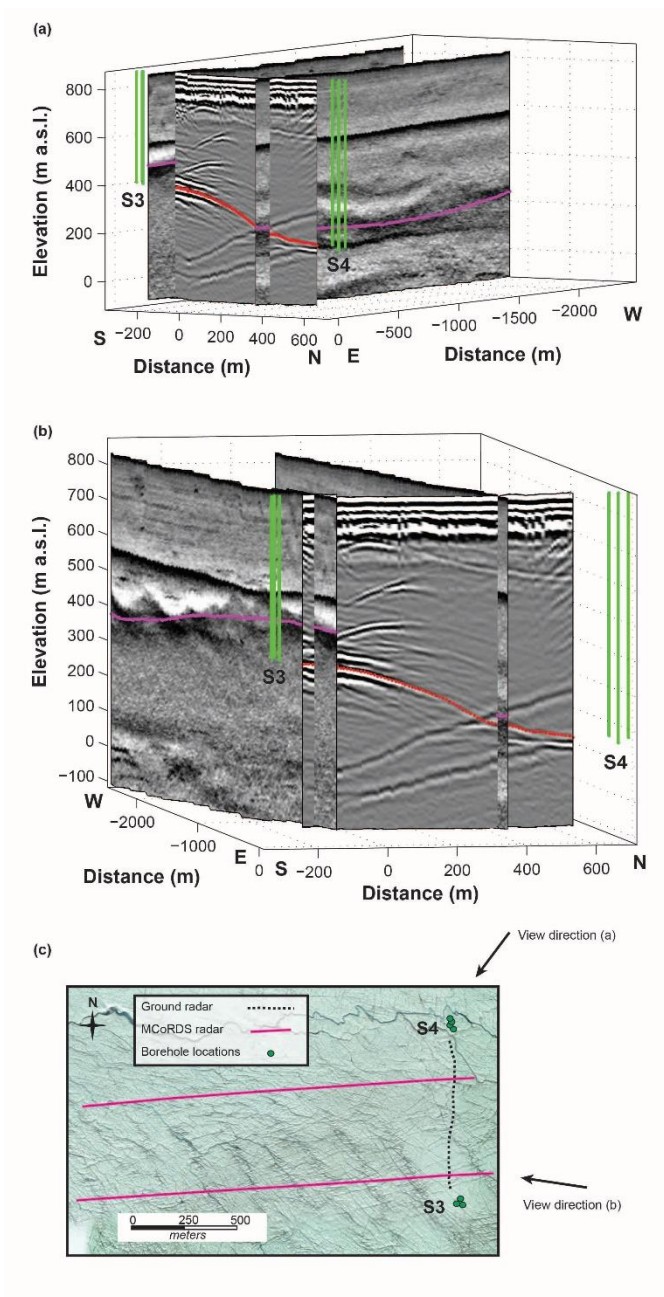

**Figure 4: Fenceplots (a) and (b) showing the 2.5 MHz radar profile and borehole location and depth (green lines) discussed in the text with two Operation IceBridge MCoRDS L1B radar echo strength profiles showing out-of-plane bed roughness. All data are hung off of the surface elevation profile of the ice. The magenta dots show the bed return interpretation from the IceBridge data, red dots show the bed return interpretation from the data presented herein. Note that the bed interpretations match well for the IceBridge data and the data presented here near S4 but the interpreted bed reflection in the IceBridge data near S3 is ~80 m above both the interpreted GPR depth to bed as well as the measured borehole depths from the data provided herein. We assume an EM propagation velocity of 1.68 x 10$^8$ m s$^{-1}$ for the GPR depth axis. (c) Map showing relative locations of radar transects in (a) and (b). Satellite image (courtesy of Google/Digital Globe).**