# Peer review of "Liquid Water Content in Ice Estimated Through a Full-Depth Ground Radar Profile and Borehole Measurements in Western Greenland"

_The Cryosphere, 2016_

## Referee Comment (RC1) · J. MacGregor (Referee) · 1 Nov 2016

J. MacGregor (Referee)

joseph.a.macgregor@nasa.gov

Summary

This manuscript describes observations and analysis of the englacial water concentration of the western Greenland Ice Sheet from two types of ground-based radar surveys and well-established methods, with validation at boreholes whose results have been described in detail in previous studies. A key finding is that the water content they infer for the temperate layer is consistently higher than that assumed by models that account for such englacial water concentrations, such that this water may have a greater influence on the flow of the ice sheet than is commonly assumed.

Major comments

This manuscript is well structured, coherently argued, well supported and well written. The data and methods are well suited to the questions posed. I find little to fault in this regard and will not belabor those points. While the observational scope of the study is fairly narrow, the broader significance of the results is reasonably established. I do not consider any of comments major.

Minor comments

15-6. While accurate, this final sentence of the abstract ought to be expanded upon in a manner consistent with both the Discussion and Conclusions sections.

2/13-14: These two sentences appear redundant in the otherwise excellent Introduction section.

3/5: Is the common-source point multi-offset survey also what's called a "walkaway" survey? If so, less of a mouthful.

4/20-29: This paragraph regarding the pros and cons of this particularly survey design seems better suited to section 2.2.

5/9-10: This sentence is surprisingly circumspect about the possibility that this reflection is due to the large englacial temperature gradient at this depth. Given the apparent coincidence between the cold–temperature transition and this reflection in Figure 3c (would be nice to also show the borehole sandwich in Figure 2), it is plausible, although the physical mechanism that generates this reflection is somewhat unclear (large increase in permittivity/conductivity/both?).

7/15-16: Not sure exactly what is meant by "grain-scale" water in cold ice. Grain-scale water bodies? Certainly liquid veins can be present even in cold ice, e.g., Dash et al. [2006, Reviews of Modern Physics, 78, doi:10.1103/RevModPhys.78.695]. This statement appears to be contradicted by later statements at the beginning of section 4.2.

8/10: Gusmeroli et al. [2010, JGR, 115, doi:10.1029/2009JF001539] is also relevant

here.

8/20: Jacobel et al. [2014, Annals of Glaciology, 55(67), doi:10.3189/AoG67A004] used low-frequency common-offset ground-based radar to study basal crevasse morphology near the grounding zone of the Siple Coast, so this statement does not seem strictly correct to me.

8/21-22: This statement is key to the broader significance of the study and I recommend expanding on it if possible.

9/7 and 10/11-12: These statements indirectly include enhanced shearing of basal/temperate ice as part of a set of "sliding processes", which doesn't sound quite right (also at odds with 9/3-4). In my perhaps conventional view, it is simpler to consider sliding processes as those that cause absolute motion of ice at the bed itself, of which I would only include direct sliding over subglacial material or mechanical failure of the latter.

Section 4.3: This section would be strengthened if it considered the bulk permittivity of ice used by CReSIS for ice thickness determination (3.15). 10/12 would especially be strengthened in this regard. The sentence about velocity assumptions in 10/11-12 is practically tautological.

9/30: While the study has elucidated the englacial water concentration of the ice sheet in this region in greater detail, it's a stretch to consider the two-layer model "complex".

10/7-8: This statement about the possible influence of an icefall should be moved earlier and shouldn't be first introduced in the Conclusions section.

Table 2 is somewhat confusing and ought to be reorganized so that upper/lower layers are shown as sub-columns of S3 and S4.

Figure 2: Add Figure 3c's borehole sandwich to each panel.

Figure 4: 3-D is tough. I appreciate the effort but had of difficulty interpreting the

different radargrams. Perhaps include a legend, a map and the view directions for each panel?

SI/S3: This section is worth including in the main text.

Grammar, etc.

1/23: that is tens of metres 3/3 and throughout manuscript: use "ice thickness" instead of "ice depth" when referring to the distance between the ice surface and bed 6/9: This agrees with the $\sim$240 m... 7/30: constant: same in time; uniform: same in space 9/9: Ryser

---

## Referee Comment (RC2) · A. Heilig (Referee) · 7 Nov 2016

The manuscript "Water Content of Greenland Ice Estimated from Ground Radar and Borehole Measurements" by Brown et al. presents a novel data set on EM wave velocities in temperate ice within the ablation zone of the Greenland Ice Sheet. The authors compare borehole records on temperature and total ice thickness with radar measurements. From conversion of two-way travel time (TWT) and measured depths, Brown et al. are able to calculate for velocities. Since measurements are conducted within the ablation zone, the contributing volume fractions of air and ice remain constant and are known. In consequence, it is possible to derive liquid water content. The estimated water contents for the temperate basal layer in this work exceed previous estimates applied in models. The presented results are of high interest to the scientific community and are worth publishing.

[Figure]

In summary, this work is relevant, well-written and adequately presented, I only ask for moderate revisions. More major points I criticize are:

- The term liquid water content (lwc) is not defined in this manuscript. It is of relevance whether you describe volumetric or gravitational lwc. Since the hoisting medium is considered as ice with a density of 917 kg/m3 (which is not defined either), you only have a ~10% offset between volumetric or gravitational lwc. However, I strongly recommend to define this term.

- The title of the manuscript is quite extensive and might lead to misinterpretations. Since you only discuss a small part within the ablation zone of an outlet glacier, I suggest to reduce the title to its geographical location.

- I agree with Joseph MacGregor that Figure 4, in the current state, is not very supportive. However, the Figure already appears in a Youtube video and consequently the video and its location must be referenced or the Fig. removed. Otherwise, this is a plagiarism!

- The chapter S3 in the supplemental information is not referenced in the manuscript. However, the information within this chapter is of high relevance and should be included into the main part.

- Furthermore, I recommend to include the error analysis of the supplement into the main part as well. This section only requires little revisions but provides important information for the reader.

Minor points that should be addressed:

- Figure 1a, the red dot within the GrIS overview is hardly visible in a print out. I suggest increasing this feature or at least provide bars leading to the respective locations.

- Figure 2 the blue and yellow dashed lines are hardly recognizable. I understand that you don't want to hide radargram information by the lines but at least consider increasing the thickness of the lines or change to colors with higher contrasts.

- I recommend to include Hobbs (Ice Physics) as reference in page 8 L30ff

---

## Referee Comment (RC3) · Anonymous Referee #3 · 21 Nov 2016

The subject of the paper: to infer water content of temperate ice from radar *and* borehole observations. After a quick read, and a careful look at the figures, I"m not sure that I understand (a) what was done, (b) how boreholes come into the paper, and (c) what was learned.

Apparently, the conclusion is buried in one of the tables, where the <10% water content is reported based on complex index of refraction analysis. I would have expected to see a graph or other similar figure that maps out the ice column between wet and dry, and which shows the principle data supporting this displayed next to the graph. The GPR images that seem to constitute the main figures are hard to understand, and the relationship to the boreholes is hard to see (why aren't the borehole data presented?).

I sense that the paper is written well for specialists in this kind of GPR remote sensing,

however as a person without this experience, I am simply lost as to what is being concluded from the presentation of the data and the analysis. I sense that the paper is overly technical, at least for my taste. I otherwise defer to people with greater expertise.

---

## Referee Comment (RC4) · Anonymous Referee #4 · 5 Dec 2016

Summary

In this study, the water content of the Greenland ice sheet near the margin is estimated. This is achieved by combining ice sheet borehole and radar surveys from a 1km long transect in southwest Greenland. The reflected travel times from the radar data are inverted to calculate the electromagnetic propagation velocity of the ice body. The borehole data are used to constrain the inversion by providing data for ice sheet depth and the boundary between temperate and cold ice. Their results points toward, higher than previously thought, water content in the ice sheet, specifically in a thick temperate ice layer right above the glacier bed. These findings are an important contribution to the study of Greenland ice sheet as it will results in improved ice viscosity estimates and modeling of ice sheet velocity fields and ice thickness.

I can't speak to the details in the radar surveys and inversion methodology, but the overall methods are well designed. The paper is for the most part easy to read and follow. I don't have any major comments. I only provide some suggestion below for the authors to consider that may clarify the manuscript to readers. In particular those readers who are interested in the topic, but does not have a foundation working with radar data would benefit from some clarification of terms.

Minor comments:

Abstract:Mention the study period (2011 to 2012?)

Page 2:L28-29: Add standard deviation to the mean values

Page 2:L26-27: Clarify how the data were retrieved (e.g. datalogger) and the timespan of the study

Page 3: L1: Add the depth of the reported temperatures and also over what time period the estimates are representative (e.g. a year?). Provide the temperature of the temperate ice. Also rephrase sentence, change "boundary with temperature ice" to something like "boundary that separates cold and temperature ice".

Page 3:L5. The meaning of "common offset" and "common source point" GPR needs to be explained for the reader. You kind of do this later, but consider briefly explaining it the first time you mention these two methods. Also, explain why it is important to use both methods.

Page 3:L9: MATLAB

Page 3:L10: Check manuscript for tense, here present tense is used " the wave is adjusted", the previous sentence used past tenste "oscilloscope was triggered". Make sure that the tense in the paper is consistent in each section.

Page 3:L24: Add reference for Ormsby bandpas filter

Page 3:L28: Explain "spatial aliasing", and "stacking" and why it matters

Page 4: L18-19: Explain "survey geometry".

Page 4: L20: Confusing. It is not clear what survey method are you using in this study, or why it matters.

Page 4: L24: Explain "dip magnitude"

Page 4: L29 Explain "Dix inversion"

Page 7: L 24: Explain "RMS misfit". Is it the same as "RMSE"?

Comments on tables and figures :

Table 1: The units are sometime provided in the header and always in the table (except for gain). Provide units in header or in the body of the table, but not both.

Table 2: Provide a header for the column between S3 and S4 (upper layer, lower layer etc). Rephrase "clear that the radar data" to "clear that the radar transect". Add text to explain that the inset Greenland map shows the study area. Add text to explain that the rectangle in panel a is the outlines of the WV-2 image in b.

Figure 2 caption: Is "reflection picks" a scientific term? Can you use another word than "picks". Also mention the geographic direction of the glacial flow in the last sentences.

Figure 3 caption: Explain "moveout", "interfacial". Clarify what "Dashed lines are the velocity model boundaries" refer to by adding the color of the dashed line in question. Explain that the temperature profiles are collected at site S3 and S4 respectively. Figure 3 other: Revise the black dashed line with another color, they are very difficult to find in panel a.

---

## Author Comment (AC1) · 27 Dec 2016

Response to: Interactive comment on "Water Content of Greenland Ice Estimated from Ground Radar and Borehole Measurements" by Joel Brown et al. J. MacGregor (Referee) joseph.a.macgregor@nasa.gov

*\*Author responses are in italics.\**

Summary

This manuscript describes observations and analysis of the englacial water concentration of the western Greenland Ice Sheet from two types of ground-based radar surveys and well-established methods, with validation at boreholes whose results have been

described in detail in previous studies. A key finding is that the water content they infer for the temperate layer is consistently higher than that assumed by models that account for such englacial water concentrations, such that this water may have a greater influence on the flow of the ice sheet than is commonly assumed.

Major comments

This manuscript is well structured, coherently argued, well supported and well written. The data and methods are well suited to the questions posed. I find little to fault in this regard and will not belabor those points. While the observational scope of the study is fairly narrow, the broader significance of the results is reasonably established. I do not consider any of comments major.

Minor comments

15-6. While accurate, this final sentence of the abstract ought to be expanded upon in a manner consistent with both the Discussion and Conclusions sections.

***We have expanded on this in the Abstract.***

2/13-14: These two sentences appear redundant in the otherwise excellent Introduction section.

***We have removed the sentence "The layer's full spatial extent can never be measured and must be addressed by modelling."***

3/5: Is the common-source point multi-offset survey also what's called a "walkaway" survey? If so, less of a mouthful.

***We agree that "common-source point multi-offset survey" is a mouthful, yet it is also quite accurate. Indeed, "walkaway" survey or "single moveout" survey also describe the survey setup, but they are subject to misinterpretation. Since***

*(1) it is our goal to clearly distinguish this survey from a CMP survey, (2) the survey setup, although not novel, is not strictly common in glaciology, and (3) it is our intention to be as clear as possible, we choose to use the most accurate description of the survey setup. We have not replaced "common-source point multi-offset survey" with another term.*

4/20-29: This paragraph regarding the pros and cons of this particularly survey design seems better suited to section 2.2.

*We have moved the paragraph as suggested by the referee.*

5/9-10: This sentence is surprisingly circumspect about the possibility that this reflection is due to the large englacial temperature gradient at this depth. Given the apparent coincidence between the cold–temperature transition and this reflection in Figure 3c (would be nice to also show the borehole sandwich in Figure 2), it is plausible, although the physical mechanism that generates this reflection is somewhat unclear (large increase in permittivity/conductivity/both?).

*This sentence is purposefully circumspect. We agree that it is possible that the temperate gradient at this boundary could coincide with a large change in permittivity or conductivity. However, without direct evidence of a large permittivity or conductivity change across this boundary from other direct measurements or from a full multi-offset survey that spans the transect, we choose not to imply that this boundary is observable in our radar profiles. Thus, we stress that this reflection is only used as a starting guess for our two-layer ray-based inversion. Optimally, a full multi-offset profile would be a better method for interpreting the water content along the profile. Unfortunately, we could not conduct a full multi-offset profile due to logistical constraints on our field campaign.*

7/15-16: Not sure exactly what is meant by "grain-scale" water in cold ice. Grain-scale water bodies? Certainly liquid veins can be present even in cold ice, e.g., Dash et al. [2006, Reviews of Modern Physics, 78, doi:10.1103/RevModPhys.78.695]. This statement appears to be contradicted by later statements at the beginning of section 4.2.

*We have clarified the term "grain-scale" water to include a reference to previous work (see next comment) which discusses microscopic water systems (water in-clusions that exist at the intersection of individual ice crystals) which are known to be present in temperate ice but are not present in ice with temperatures below the pressure melting point of ice.*

8/10: Gusmeroli et al. [2010, JGR, 115, doi:10.1029/2009JF001539] is also relevant here.

*We have added the reference here. We have also added e.g. to the beginning of the list of references here to indicate that the list is not exhaustive.*

8/20: Jacobel et al. [2014, Annals of Glaciology, 55(67), doi:10.3189/AoG67A004] used low-frequency common-offset ground-based radar to study basal crevasse mor-phology near the grounding zone of the Siple Coast, so this statement does not seem strictly correct to me.

*The common offset radar used to detect basal crevassing in the Jacobel et al. study were collected in TM mode, meaning the radar were collected with the antennae laid out end-to-end. As they describe in their paper, the antennae beam pattern from TM mode can result in off-nadir reflections that are stronger than nadir reflections. In their study, Jacobel et al. inferred the presence of basal crevasses from these off-nadir reflections. Our common-offset data were collected in TE mode which typically results in the strongest reflections*

[Figure]

*occurring from nadir or near nadir. We have not revised the sentence since our methods were not targeted to their detection.*

8/21-22: This statement is key to the broader significance of the study and I recommend expanding on it if possible.

*We agree that this is key to the broader significance of the study. We expand on this significance in Section 4.2 and again in the last sentence in the conclusion. We are unaware of studies that model the high strain rates implied by the high wetness values for the temperate layer shown in this study, as we state in the manuscript, there is typically a 1We have not expanded further upon this key point.*

9/7 and 10/11-12: These statements indirectly include enhanced shearing of basal/temperate ice as part of a set of "sliding processes", which doesn't sound quite right (also at odds with 9/3-4). In my perhaps conventional view, it is simpler to consider sliding processes as those that cause absolute motion of ice at the bed itself, of which I would only include direct sliding over subglacial material or mechanical failure of the latter.

*9/7 - We have changed "thus making the distinction between high straining of the temperate layer and other sliding processes ambiguous." To "thus making the distinction between motion due to high straining of the temperate layer and motion due to sliding processes ambiguous." 10/11-12 – We have removed the last line of the manuscript which the referee points out is practically tautological (below).*

Section 4.3: This section would be strengthened if it considered the bulk permittivity of ice used by CReSIS for ice thickness determination (3.15). 10/12 would especially be

[Figure]

strengthened in this regard. The sentence about velocity assumptions in 10/11-12 is practically tautological.

*We added the information as suggested by the reviewer. However, we would like to point out that the CReSIS provided depth scale for the data use a permittivity of 1 to convert from traveltime to depth. We assume that this is NOT what is used to determine depth of basal reflectors and that a bulk permittivity of 3.15 is used to determine ice thickness. We have removed the last line of the manuscript which the referee points out is practically tautological.*

9/30: While the study has elucidated the englacial water concentration of the ice sheet in this region in greater detail, it's a stretch to consider the two-layer model "complex".

*We removed the word 'complex' the sentence now reads: "Our integration of ground based-radar data with information collected in boreholes reveals a two-layer, thermo-hydrologic structure of varying thicknesses in the ablation zone of western Greenland."*

10/7-8: This statement about the possible influence of an icefall should be moved earlier and shouldn't be first introduced in the Conclusions section.

*We respectfully disagree with the reviewer on this point. The provenance of water-filled voids in this section is speculation and does not merit further comment or attention in the manuscript. We have not moved the location of the statement.*

Table 2 is somewhat confusing and ought to be reorganized so that upper/lower layers are shown as sub-columns of S3 and S4.

*We have reorganized the Table as the reviewer suggested.*

Figure 2: Add Figure 3c's borehole sandwich to each panel.

***The borehole data has a vertical axis of depth; the panels of Figure 2 have a vertical axis of traveltime. This was done on purpose to show the data before the depth conversion was made. The depth conversion in Figure 3c (as we state in the figure caption) were calculated from the final velocity model shown in Figure 3c. Using the final velocity model to convert the data from TWT to depth in Figure 2 is not appropriate as this data was used in the derivation of the velocity model. We did not add the requested data to the other panels of this figure.***

Figure 4: 3-D is tough. I appreciate the effort but had of difficulty interpreting the different radargrams. Perhaps include a legend, a map and the view directions for each panel?

***We have added a map view with the relative locations of the three radar transects to this figure.***

SI/S3: This section is worth including in the main text.

***We have moved this section to the main text as suggested by two reviewers.***

Grammar, etc. 1/23: that is tens of metres

3/3 and throughout manuscript: use "ice thickness" instead of "ice depth" when referring to the distance between the ice surface and bed

6/9: This agrees with the 240 m. . .

7/30: constant: same in time; uniform: same in space

9/9: Ryser

***We have made all of the grammar changes suggested by the referee***

---

## Author Comment (AC2) · 27 Dec 2016

Response to: Interactive comment on "Water Content of Greenland Ice Estimated from Ground Radar and Borehole Measurements" by Joel Brown et al. A. Heilig (Referee) heilig@r-hm.de

*Author responses are in italics.*

The manuscript "Water Content of Greenland Ice Estimated from Ground Radar and Borehole Measurements" by Brown et al. presents a novel data set on EM wave velocities in temperate ice within the ablation zone of the Greenland Ice Sheet. The authors compare borehole records on temperature and total ice thickness with radar measure-
ments. From conversion of two-way travel time (TWT) and measured depths, Brown et al. are able to calculate for velocities. Since measurements are conducted within the ablation zone, the contributing volume fractions of air and ice remain constant and are known. In consequence, it is possible to derive liquid water content. The estimated water contents for the temperate basal layer in this work exceed previous estimates applied in models. The presented results are of high interest to the scientific community and are worth publishing.

In summary, this work is relevant, well-written and adequately presented, I only ask for moderate revisions. More major points I criticize are:

The term liquid water content (lwc) is not defined in this manuscript. It is of relevance whether you describe volumetric or gravitational lwc. Since the hoisting medium is considered as ice with a density of 917 kg/m3 (which is not defined either), you only have a 10

*Because the CRIM and Looyenga equations are explicit in solving for the volumetric percentage of water the, calculated liquid water content is volumetric. We have added this information to the manuscript.*

The title of the manuscript is quite extensive and might lead to misinterpretations. Since you only discuss a small part within the ablation zone of an outlet glacier, I suggest to reduce the title to its geographical location.

*We have changed the title to "Liquid Water Content in Ice Estimated Through a Full-Depth Ground Radar Profile and Borehole Measurements in Western Greenland"*

I agree with Joseph MacGregor that Figure 4, in the current state, is not very supportive. However, the Figure already appears in a Youtube video and consequently the

video and its location must be referenced or the Fig. removed. Otherwise, this is a plagiarism!

*We have added the video, which we made, to the supplementary information. We have not cited the YouTube video as it is not a scientific source, it was uploaded for display on a personal website of one of the authors.*

The chapter S3 in the supplemental information is not referenced in the manuscript. However, the information within this chapter is of high relevance and should be included into the main part.

*We have moved this information to the main text since two referees suggested that the information is highly relevant.*

Furthermore, I recommend to include the error analysis of the supplement into the main part as well. This section only requires little revisions but provides important information for the reader.

*We agree that the error analysis is important for understanding the limitations of this study. However, we feel that including the description of how the errors were estimated to the main text, distracts from the continuity of the paper. This section is included in the Supplementary information for readers who are interested in the error estimation. We did not move this section to the main text.*

Minor points that should be addressed: Figure 1a, the red dot within the GrIS overview is hardly visible in a print out. I suggest increasing this feature or at least provide bars leading to the respective locations.

*We have added an arrow pointing to the location in Figure1a.*

Figure 2 the blue and yellow dashed lines are hardly recognizable. I understand that you don't want to hide radargram information by the lines but at least consider increasing the thickness of the lines or change to colors with higher contrasts.

*We have experimented with many iterations of the appropriate weight and color of these lines. We find that the increasing the weight of the lines completely masks the data we are trying to highlight, the same is true for changing the color of the lines. We have not changed the color or weight of these lines.*

I recommend to include Hobbs (Ice Physics) as reference in page 8 L30ff

*Although Hobbs' book is informative, since (1) the majority of the book does not deal with water inclusions and (2) the book was published after the two citations included in this line, we do not cite Hobbs. We have added e.g. to this list of references to show that the list is not exhaustive.*

––––––––––––––––––––––––––––

---

## Author Comment (AC3) · 27 Dec 2016

*\*Author responses are in italics.\**

The subject of the paper: to infer water content of temperate ice from radar *and* borehole observations. After a quick read, and a careful look at the figures, I"m not

sure that I understand (a) what was done, (b) how boreholes come into the paper, and (c) what was learned. Apparently, the conclusion is buried in one of the tables, where the <10% water content is reported based on complex index of refraction analysis. I would have expected to see a graph or other similar figure that maps out the ice column between wet and dry, and which shows the principle data supporting this displayed next to the graph. The GPR images that seem to constitute the main figures are hard to understand, and the relationship to the boreholes is hard to see (why aren't the borehole data presented?).

***Borehole temperature data are presented in Figure 3c and borehole depth data are presented in the text of Section 2.1.***

I sense that the paper is written well for specialists in this kind of GPR remote sensing, however as a person without this experience, I am simply lost as to what is being concluded from the presentation of the data and the analysis. I sense that the paper is overly technical, at least for my taste. I otherwise defer to people with greater expertise.

***This reviewer states that he/she lacks the scientific background to understand this research. The reviewer provides no criticisms or suggestions that can be addressed, and defers to experts in the field. We have taken no action in response to this review.***

---

## Author Comment (AC4) · 27 Dec 2016

Response to: Interactive comment on "Water Content of Greenland Ice Estimated from Ground Radar and Borehole Measurements" by Joel Brown et al.

Anonymous Referee 4

*\*Author responses are in italics.\**

Summary

In this study, the water content of the Greenland ice sheet near the margin is estimated.

This is achieved by combining ice sheet borehole and radar surveys from a 1km long transect in southwest Greenland. The reflected travel times from the radar data are inverted to calculate the electromagnetic propagation velocity of the ice body. The borehole data are used to constrain the inversion by providing data for ice sheet depth and the boundary between temperate and cold ice. Their results points toward, higher than previously thought, water content in the ice sheet, specifically in a thick temperate ice layer right above the glacier bed. These findings are an important contribution to the study of Greenland ice sheet as it will results in improved ice viscosity estimates and modeling of ice sheet velocity fields and ice thickness.

I can't speak to the details in the radar surveys and inversion methodology, but the overall methods are well designed. The paper is for the most part easy to read and follow. I don't have any major comments. I only provide some suggestion below for the authors to consider that may clarify the manuscript to readers. In particular those readers who are interested in the topic, but does not have a foundation working with radar data would benefit from some clarification of terms.

*Author note: We understand that 'The Cryosphere' is intended for a scientific audience with a broad range of specialties in the cryospheric sciences and that some technical jargon used in this paper may be unfamiliar to part of the audience of the journal. However, this is true for every scientific paper published in this journal. In these comments from referee 4 there are many requests to explain basic geophysical terms, specifically comments 7, 10, 11, 14, 15, 16, 19, and 20 (below). We agree with the referee that a 'foundation working with radar data' greatly improves the potential for understanding the exact methods used in this study and appreciate the suggestions of the referee for increasing the understanding of the manuscript for a broader audience. However, we do not believe that the main text of the manuscript is a proper place for defining geophysical terminology. With this in mind, we have added a short glossary*

*of selected terminology to the Supplementary Information with the knowledge that in-depth understanding of the geophysical terminology will require further study by the reader lacking a foundation working with radar.*

Minor comments:

1. Abstract:Mention the study period (2011 to 2012?)

*We have added this information to the Abstract.*

2. Page 2:L28-29: Add standard deviation to the mean values

*We have added the standard deviation to the mean values*

3. Page 2:L26-27: Clarify how the data were retrieved (e.g. datalogger) and the timespan of the study

*We have added how the data were retrieved.*

4. Page 3: L1: Add the depth of the reported temperatures and also over what time period the estimates are representative (e.g. a year?).

*We have added the depth of the temperatures and that the temperatures are representative of the time over which they were recorded.*

5. Provide the temperature of the temperate ice.

*Temperate ice is, by definition, at the pressure melting point of ice, which changes with depth. This approximated relationship is given in figure 3c. We have made no changes in response to this comment.*

6. Also rephrase sentence, change "boundary with temperature ice" to something like "boundary that separates cold and temperature ice".

*We have changed the sentence as suggested by the reviewer.*

7. Page 3:L5. The meaning of "common offset" and "common source point" GPR needs to be explained for the reader. You kind of do this later, but consider briefly explaining it the first time you mention these two methods. Also, explain why it is important to use both methods.

*"Common offset" is very descriptive and precise terminology for GPR surveys; "Common source point" is less common terminology for GPR surveys in glaciology, however, it is out of the scope of this paper to define basic GPR terminology. Further, as the reviewer states, we describe the acquisition setup for each method in the third and final paragraph in this section (2.2). To avoid repetition, we did not add to this section. Please refer to the 'Author note' above.*

8. Page 3:L9: MATLAB

*We have changed the case of the lettering as the reviewer suggests.*

9. Page 3:L10: Check manuscript for tense, here present tense is used " the wave is adjusted", the previous sentence used past tenste "oscilloscope was triggered". Make sure that the tense in the paper is consistent in each section.

*The tense in the manuscript is consistent. All sentences that deal with data processing are in present tense. All sentences that deal with data acquisition are in past tense. This is the proper way of presenting each.*

10. Page 3:L24: Add reference for Ormsby bandpass filter

**This is a commonly used filter in geophysics. Please refer to the 'Author note' above.**

11. Page 3:L28: Explain "spatial aliasing", and "stacking" and why it matters

**Please refer to the 'Author note' above.**

12. Page 4: L18-19: Explain "survey geometry".

**We have replaced "This survey geometry is…" with "Common-source point multi-offset surveys are…" in this sentence.**

13. Page 4: L20: Confusing. It is not clear what survey method are you using in this study, or why it matters.

**This section describes why the survey that we use is more appropriate for our field site than CMP surveys. We have rewritten the section for clarity of which survey we are using.**

14. Page 4: L24: Explain "dip magnitude"

**Dip angle is a basic, common geological term. Please refer to the 'Author note' above. We have changed "dip magnitude" to "the magnitude of the dip angle" for clarity.**

15. Page 4: L29 Explain "Dix inversion"

**The Dix inversion is a common calculation used in geophysics to solve for the**

**layer by layer properties of the subsurface. Please refer to the 'Author note' above. We have added a reference to the dix inversion for clarity.**

16. Page 7: L 24: Explain "RMS misfit". Is it the same as "RMSE"?

**This is not the same as RMSE (RMS error). The RMS misfit is not a measurement of error, it is a measurement of how closely the forward model traveltime results match the multi-offset data. Please refer to the 'Author note' above.**

Comments on tables and figures:

17. Table 1: The units are sometime provided in the header and always in the table (except for gain). Provide units in header or in the body of the table, but not both.

**Gain does not have a unit, it is a coefficient. Please refer to the 'Author note' above. We have moved all units to the header.**

18. Table 2: Provide a header for the column between S3 and S4 (upper layer, lower layer etc). Rephrase "clear that the radar data" to "clear that the radar transect". Add text to explain that the inset Greenland map shows the study area. Add text to explain that the rectangle in panel a is the outlines of the WV-2 image in b.

**We have made the changes that the reviewer suggests.**

19. Figure 2 caption: Is "reflection picks" a scientific term? Can you use another word than "picks". Also mention the geographic direction of the glacial flow in the last sentences.

**Reflection 'picks' is a scientific (and descriptive) term; thus, it is the appropriate term to use here. Please refer to the 'Author note' above. We have added the**

*geographic flow direction to the caption.*

20. Figure 3 caption: Explain "moveout", "interfacial".

***These are basic geophysical terms and therefore out of the scope of description of this paper. Please refer to the 'Author note' above.***

21. Clarify what "Dashed lines are the velocity model boundaries" refer to by adding the color of the dashed line in question.

***We have clarified that the dashed lines that represent the model boundary are in panel (b).***

22. Explain that the temperature profiles are collected at site S3 and S4 respectively.

***We have labeled the temperature profiles with their respective locations.***

23. Figure 3 other: Revise the black dashed line with another color, they are very difficult to find in panel a.

***We have changed the color of the dashed lines in panel a to red.***